# The Role of Exosomes in Stemness and Neurodegenerative Diseases—Chemoresistant-Cancer Therapeutics and Phytochemicals

**DOI:** 10.3390/ijms21186818

**Published:** 2020-09-17

**Authors:** Narasimha M. Beeraka, Shalini H. Doreswamy, Surya P. Sadhu, Asha Srinivasan, Rajeswara Rao Pragada, SubbaRao V. Madhunapantula, Gjumrakch Aliev

**Affiliations:** 1Center of Excellence in Regenerative Medicine and Molecular Biology (CERM), Department of Biochemistry, JSS Medical College, JSS Academy of Higher Education & Research (JSS AHER), Mysuru 570015, Karnataka, India; bnmurthy24@gmail.com (N.M.B.); shalini.hd98@gmail.com (S.H.D.); 2AU College of Pharmaceutical Sciences, Andhra University, Visakhapatnam 530003, Andhra Pradesh, India; sprabha125@gmail.com (S.P.S.); profprrau@gmail.com (R.R.P.); 3Center of Excellence in Regenerative Medicine and Molecular Biology (CERM), Division of Nanoscience and Technology, Faculty of Life Sciences, JSS Academy of Higher Education & Research (JSS AHER), Mysuru 570015, Karnataka, India; asha.srinivasan@jssuni.edu.in; 4Center of Excellence in Molecular Biology and Regenerative Medicine (CEMR), Department of Biochemistry, JSS Medical College, JSS Academy of Higher Education & Research (JSS AHER), Mysuru 570015, Karnataka, India; 5Special Interest Group in Cancer Biology and Cancer Stem Cells (SIG-CBCSC), JSS Medical College, JSS Academy of Higher Education & Research (JSS AHER), Mysuru 570015, Karnataka, India; 6Sechenov First Moscow State Medical University (Sechenov University), St. Trubetskaya, 8, bld. 2, 119991 Moscow, Russia; 7Institute of Physiologically Active Compounds, Russian Academy of Sciences, 142432 Chernogolovka, Moscow Region, Russia; 8Research Institute of Human Morphology, 3 Tsyurupy Street, 117418 Moscow, Russia; 9GALLY International Research Institute, 7733 Louis Pasteur Drive, #330, San Antonio, TX 78229, USA

**Keywords:** exosomes, cancers, miRNAs, lncRNAs, stemness, chemo-/radio-resistance, angiogenesis, metastasis, neurodegeneration, phytochemicals

## Abstract

Exosomes exhibit a wide range of biological properties and functions in the living organisms. They are nanometric vehicles and used for delivering drugs, as they are biocompatible and minimally immunogenic. Exosomal secretions derived from cancer cells contribute to metastasis, immortality, angiogenesis, tissue invasion, stemness and chemo/radio-resistance. Exosome-derived microRNAs (miRNAs) and long non-coding RNAs (lnc RNAs) are involved in the pathophysiology of cancers and neurodegenerative diseases. For instance, exosomes derived from mesenchymal stromal cells, astrocytes, macrophages, and acute myeloid leukemia (AML) cells are involved in the cancer progression and stemness as they induce chemotherapeutic drug resistance in several cancer cells. This review covered the recent research advances in understanding the role of exosomes in cancer progression, metastasis, angiogenesis, stemness and drug resistance by illustrating the modulatory effects of exosomal cargo (ex. miRNA, lncRNAs, etc.) on cell signaling pathways involved in cancer progression and cancer stem cell growth and development. Recent reports have implicated exosomes even in the treatment of several cancers. For instance, exosomes-loaded with novel anti-cancer drugs such as phytochemicals, tumor-targeting proteins, anticancer peptides, nucleic acids are known to interfere with drug resistance pathways in several cancer cell lines. In addition, this review depicted the need to develop exosome-based novel diagnostic biomarkers for early detection of cancers and neurodegenerative disease. Furthermore, the role of exosomes in stroke and oxidative stress-mediated neurodegenerative diseases including Alzheimer’s disease (AD), and Parkinson’s disease (PD) is also discussed in this article.

## 1. Introduction

### Extracellular Vesicles

The research work on extracellular vesicles (EVs) is experiencing an unprecedented spurt in a decade in the field of cancer and neurodegenerative diseases [1,2]. EVs are a heterogeneous population of secreted submicron-size microparticles and the nanometer-size exosomes involved in cell-to-cell communication. For instance, the delivery of cargo such as multi-drug resistance (MDR)-associated proteins and miRNAs through exosomes to the bystander cancer cells induces drug resistance during chemotherapy [1,3,4]. EVs based on their origin, size and mode of release are categorized into exosomes (30–100 nm), microvesicles (50–1000 nm), and apoptotic bodies (50–5000 nm) [5].

Exosomes are tiny vesicles epitomized by the presence of a stable membranous covering, enclosing several molecules that include immune components, hormones, sugars, steroids, RNAs, microRNAs, lipids, and nucleic acid polymers, etc. [6,7,8]. Exosomes are generated from both normal and cancerous cells inside the tissues [1]. The exosomal constituents vary significantly depending on the specific traits of parent cells from which they are originated [9]. In addition, the content of an exosome is an indicative of metabolic status of their originating cells, and, hence, can be considered as a fingerprint of the originating cells. Furthermore, the significance of exosomes can lay a pavement for cancer biologists and neurobiologists to efficiently understand underlying complexities in cancers [1] and detrimental oxidative-stress related neurodegenerative disorders [2].

Exosomes are reported to be involved in mediating the uncontrolled proliferation of cancer cells, invasion, metastasis, chemoresistance, stemness [10,11], tumor aggressiveness [12], and exhibit an impact in modulating the tumor microenvironment [13]. Composition of exosomes secreted from cancer cells denotes the state of tumor as well as tumor microenvironment [14]. The versatility, stability, and ubiquitous nature of exosomes make them ideal carriers of cargo [7]. Hence, exosomes can be efficiently explored for developing novel therapeutic interventions against cancer and oxidative stress-related neurodegenerative diseases [7,15]. The inner cargo of exosomes confer several cellular and physiological functions as these components are involved in cell signaling to regulate phenotypic alterations through the reprogramming of genes, metabolic processes, and cell signaling pathways in recipient cells [16,17].

Exosomes production inside the cells occurs through exosome biogenesis; however, the content sorting into exosomes is predominantly regulated by cellular and molecular mechanisms [7]. The generation of exosomes is a complex process, which involves the invagination of endosomal membrane into early endosomes followed by the formation of multivesicular bodies (MVBs) (Figure 1) [18]. Exosomes are developed, in general, from pre-exosomes that contain DNA, RNA, and proteins [7]. Pre-exosomes undergo specific sorting mechanisms including segregation of different cargo to form a mature exosome [7]. The specific sorting of cargo into the exosomes is significantly controlled by several cell signaling mechanisms viz., endosomal sorting complexes required for transport (ESCRT-0, -I, -II, and -III), tetraspanins, [19,20], light-dependent signaling, etc. [21,22]. Content sorting into exosomes is also regulated by the zipcode of 3′-UTR in exosomal mRNAs [21]. Specific GGAG motifs foster the explicit sorting of miRNA cargo into exosomes by interacting with specific chaperone proteins [23].

Several cellular model systems are reported to be involved in regulating transportation of MVBs to the plasma membrane [24,25,26,27]. For instance, the docking of MVBs to the plasma membrane and secretions of exosomes into outer-cellular milieu could be predominantly influenced by the activity of Rab-GTPases, neutral sphyngomyelinase-2, and SNARE complexes [13,18,28,29,30]. Thus, the selectivity of cargo into exosomes and their modulation could be used to develop exosome-based targeted therapeutics.

Moreover, the uptake of exosomes by recipient cells elicits specific exosomal functions for cell-to-cell communication [7,8]. The process of exosomal uptake is initiated by the activity of specific recipient cell surface receptors [31,32]; where the exosomes undergo internalization either by endocytosis or by direct fusion, followed by releasing its internal cargo into the recipient cell. A recent study [33] described the blockade of exosomal uptake by the recipient cells if the cancer cells are subjected to proteinase-K treatment [33]. Hence, the development of novel therapeutic interventions to target this binding process between exosomes and recipient cells could be a promising approach against cancer.

Exosomes are involved in immune regulation in newborns since they can be found in human breast milk [7,15]. In addition, exosomes confer intestinal epithelial growth and prevent the incidence of necrotizing enterocolitis [34,35,36,37]. Exosomes are involved in maintaining the functional aspects of male/female reproductive urogenital tracts and simultaneously maintain placental and fetal health to enhance the healthy pregnancy [38]. Breast milk exosomes foster the efficient passage of microRNAs from mother to infant and actuate immunogenic health [38]. Tolerising exosomes, also referred to as “tolerosomes”, which are present in human breast milk, are reported to suppress allergic responses [16,39,40,41,42]. Biomolecular profiling of exosomes enabled the development of “Exocarta”, the largest exosome content database [http://www.exocarta.org], which describes the total number of proteins (41,860), lipids (1116), RNAs including miRNAs, noncoding RNAs (7540) identified in exosomes of several multicellular organisms [43,44,45,46,47,48,49,50,51]. Specifically, EV-linked short noncoding RNAs, miRNAs, and lncRNAs isolated from either blood or urine of cancer patients may be considered as early diagnostic and prognostic markers for specific cancer types. For instance, miR-21 is referred as one of the significant serum or plasma cancer diagnostic markers for various cancers [52]. EV-ncRNAs may serve as bonafide signatures for tumor recurrence and overall survival, as these signatures have significant prognostic implications against chemotherapy, and radiotherapy against multiple cancers [52].

Exosome secretions from cellular sources offer potential therapeutic tools against both cancers and neurodegenerative diseases, as they exhibit transport capacity of therapeutic entities against brain tumors, and neurodegenerative diseases. Furthermore, exosomes can facilitate the delivery of phytochemicals and other drugs to even cross the blood brain barrier [53]. Unique and sequestered cargos in exosomes represent the physiological health or pathophysiological state of originating cell source in cancers (for ex. stemness), and neurodegenerative diseases. The targeted delivery of therapeutic proteins loaded into EVs in preclinical models proved their efficacy against neurodegenerative diseases (Parkinson’s disease), Schwannomas, and gliomas [53]. For instance, packaging of suitable mRNA into EVs to code prodrug activating enzymes (i.e., suicide proteins) could induce Schwannomas regression. Suicide gene is composed of cDNA coding for two enzymes viz., cytosine deaminase (CD) and uracil phosphoribosyltransferase (UPRT) to mediate the conversion of prodrug of 5-fluorocytosine (5-FC) in a synergistic manner into 5-FU, subsequently into 5-fluoro-deoxyuridine monophosphate (5-FdUMP), which ultimately released across the tumor microenvironment [53,54,55]. https://www.ncbi.nlm.nih.gov/pmc/articles/PMC4860146/—R48 This active chemotherapeutic molecule can block DNA synthesis and initiate cell death. Therefore, therapeutic modalities similar to this can be designed using EVs to mitigate the activity of factors causing neuropathology in gliomas, and neurodegenerative diseases.

In addition, exosomes are known to play a significant role in aiding novel biosensing systems to ascertain the targeted delivery of therapeutic entities across the tumor microenvironment, and affected brain regions during neurodegeneration. EV-enriched with survivin has been implicated in situ applications as a novel biosensing system to equip/evaluate nanocarriers of controlled drug delivery systems against chemoresistant cancers. Hence, the analysis of survivin biomarkers in exosomes and tumor cells can be used as non-invasive liquid biopsy [56]. The present review is a promising platform for developing diagnostic and prognostic biomarkers for early detection of cancer and neurodegeneration based on the cargo inside the EVs. Moreover, this review also provides an overview of repurposing EVs as carriers of therapeutics to target various chemoresistant cancers including brain tumors, and neurodegenerative diseases. Furthermore, the global profiling of EV-RNAs against cancer mutations, stemness, disease prognosis, tumor recurrence and overall survival is a significant strategy to predict cancer specific signatures and offer a platform to develop novel RNA-based EVs as disease biomarkers.

## 2. Exosomes and Stemness

Exosomes are characterized by the presence of specific receptors distributed in the outer lipid layer [16,57,58,59]. Exosomes are composed of cellular cargo inside and are usually differentiated by size and specific surface markers viz., “TSG101, Alix, Flotillin-1 CD63, and CD9” [23,60,61,62,63]. Exosomes stimulate a variety of target cells by releasing their inner cargo [64,65]. Several reports have demonstrated that extracellular vesicles and exosomes play a significant role in cancer progression, and tumorigenesis [66,67,68]. For instance, the exosomes generated from mesenchymal stromal cells (MSCs) or fibroblasts impart cancer cells to acquire uncontrolled proliferative ability and chemoresistance as observed in multiple myeloma, colon cancer, and gastric cancers. These exosomes deliver several microRNAs and soluble factors into adjacent tumor cells [11,69,70,71,72]. EV-linked lncRNAs can foster differential patterns of epigenetic regulation, cellular reprogramming, and genomic instability in recipient cells, which are ultimately conducive to the cancer initiating signatures, and chemo-/radio-resistance [52].

Astrocyte-derived exosomes are involved in the transfer of miR-17~92 clusters in order to block the activity of PTEN genes in brain tumors [72]. In addition, the malignant cancer cells could generate a vast amount of exosomes to mediate uncontrolled proliferation of endothelial cells and angiogenesis, which further fosters tumor progression through metastasis [73,74]. Exosomes released from the cancer cells mediate immunosuppression by modulating the activity of TAMs in the tumor microenvironment; which further promotes tumor survival, growth, and chemoresistance as this exosomal cargo could foster the chemoresistance signaling in cancer cell [74,75,76,77]. AML cell-derived exosomes can release miR-155 into the human “normal hematopoietic stem and progenitor cells (HSPCs)” consequently blocking the expression of c-Myb, alleviating hematopoiesis and provoking leukemia cell proliferation [78]. Exosomes not only contribute to the development of cancerous cells from normal epithelial cells, but also to invade extracellular matrix and distal metastasis [79,80,81]. The protein content of exosome-derived cargo in the tumor cells has a significant influence on the organotropism and non-random patterns of metastasis [69]. Exosomes are reported to be involved in conferring epithelial-to-mesenchymal transition (EMT) in benign/malignant breast cancer via TGF-β2 upregulation [82]. A recent report by Qadir et al. (2018) showed the transcriptome reprogramming of exosomes in HNOK oral cancer cell lines [83]. The cancer-derived exosomes could modulate (a) the matrix remodelling factors MMP-9, EFEMP1, DKK3, SPARC; (b) cytoskeletal proteins viz., “TUBB6, FEZ1, CCT6A”, (c) deubiquitin factors, (d) membrane trafficking factors, (e) apoptosis, (f) cell cycle, and (g) transcription/translation factors that contribute to the angiogenesis, metastasis, and immune evasion in cancers [83]. Exosome-mediated transcriptome reprogramming requires extensive preclinical and clinical evaluation to develop novel liquid-biopsy based diagnostics and immunotherapies against chemoresistant cancers. However, it is imperative to know whether cancer cells exploit exosomes in modulating these factors that are responsible for chemoresistance using transcriptome reprogramming [83].

Radiation therapy (RT) is one of the frequently used options for treating tumors, especially against breast cancers. RT involves irradiation of high-energy rays to inhibit tumor cell growth. The radiated tumor cells secrete exosomal cargo to mediate cell-to-cell communication during radiation therapy, and consequently acquire therapeutic resistance to IR irradiation [84]. In addition, the exosomes from stressed and radiated cells could modify adjacent cell function and enhance the bystander effect, i.e., stemness on the adjacent cells to become more radioresistant and foster tumor aggressiveness [4,85]. For instance, the irradiation of MCF-7 cells in vitro with X-rays significantly conferred exosome biogenesis and radioresistance in a dose-dependent manner [84]. The exosomes-mediated chemo-/radio-resistance have been extensively discussed in the following sections.

## 3. Exosomes and Neurodegeneration

Exosome secretions can be observed in neurons, astrocytes, oligodendrocytes, microglia, and neural stem cells, and is reported to play a vital role in several neurological diseases [86,87]. Exosomes are involved in health and diseases, since they play a prominent role in several cellular functions including immune modulation, cell-to-cell signaling, stem cell proliferation (in particular during chemoresistant gliomas), neuronal function, and viral replication [88,89]. Exosome biochemistry has a significant influence on the neurodegeneration while disease-derived exosomes propagate to healthy cells including neuronal cells simultaneously infecting both distant and neighbouring cells [90]. Bellingham et al. (2015) have described the involvement of exosomes in metal homeostasis including neurodegeneration [2]. Moreover, exosomes can be used as vehicles for the nanodrug delivery to deliver drug molecules across the blood-brain barrier (BBB) against neurodegenerative diseases [91]. A recent review by Tarasov et al. (2019) described the efficiency of exosomes for the selective nanodrug delivery of drug molecules/gene-therapeutics/immunotherapeutics across BBB [20]. However, the drawbacks of using exosomes for nanodrug delivery to promote effective neuronal delivery for treating neurodegneration include (a) difficulties associated with isolation and purification; (b) limited quantity; (c) and expensive, especially if intended to test in vivo models.

## 4. Role of Exosomes in the Invasion and Metastasis

Invasion and metastasis are progressive pathogenic features of cancers [92]. The biological and genetic changes of dividing cancer cells are transferred to the exosomes after several proliferation cycles, finally leading to the sharing of biological characteristics among cancer cells [92]. Metastasis is a process of cancer cell invasion via blood/lymph flow and undergoes colonization in distant organs to develop tumor growth. Several cell-signaling events inside the tumor microenvironment induce invasion and metastasis. Tumor exosomes are involved in influencing several cell-signaling cascades responsible for both invasion and metastasis of cancer cells. For instance, exosomes are considered to be significant signal transfer vehicles and vectors for carrying genetic information, concluding that targeting exosomes can be an effective approach against cancers. A report by Al-Nedawi et al. (2008) has described the ability of exosomes for intercellular transfer of the mutated-oncogenic EGFR VIII from glioma cells [93]. One more report elucidated the exosomal transport of mutant oncogenic DKO-1 (mutant KRAS), and wild type DKs-8 (KRAS allele) sequentially enhanced the invasion, metastasis, and growth of colon tumors in a three-dimensional manner as mutant KRAS colon cells composed of several tumor-promoting proteins, viz., KRAS, EGFR, SRC, and integrins [94,95]. Small GTPases RAB27A/B and SMPD3/NSMASE2 are the proteins which can regulate the formation of MVBs into exosomes and exosomal miRNA secretion [96]. Rab27a/b or nSMase2 involved in the cancer cell metastasis via miRNA-mediated exosomal pathways [28,96]. Hence, the strategies to block the exosomal release of miRNAs by the knockdown of Rab27a/b or nSMase2 genes could inhibit cancer cell metastasis and tumor invasion [14]. In addition, the exosomes are transferred from malignant cancer cells to benign and less aggressive tumor cells and aid in uncontrolled proliferation, invasion and metastasis [97]. The miR-200-containing exosomes significantly aid in breast cancer cell metastasis and mediate colonization at distant sites [98]. It was reported that the uptake of the extracellular vesicles as exosomes could be attributed to the transfer of metastatic ability in breast cancer cells [98].

A plethora of research reports describe the role of TAMs in the tumor microenvironment to promote the cancer cell invasion and metastasis via exosome-dependent pathways. For instance, TAMs can release the exosomes loaded with functional ApoE (Apolipoprotein E) to the cancer cells in tumor microenvironment and enhance the metastatic spread of gastric cancer cells [99]. Another report by Jingqin Lan et al. (2019), described the role of TAMs-derived miRNA-loaded exosomes in modulating metastasis and invasion of CRC cells [92]. Exosomes containing several miroRNAs could modulate the polarization of TAMs into M2 phenotype, which further enhance the uncontrolled proliferation, invasion, and metastasis of ovarian cancer cells [100,101,102,103]. Thus, exosomes, TAMs, and tumor microenvironments have a significant close relation in mediating invasion and metastasis. Cancer cells also have the ability in modulating microenvironment of the distant organs using exosome signaling pathways to facilitate metastasis [104]. Nanovesicles could influence the T cell immune functions and skew the innate immune cells to polarize towards the protumorigenic phenotype [105]. These functions contribute to the delivery of molecular signals for promoting neovascularization and metastasis through exosomes [105].

Exosomes loaded with integrins can play a significant role in the organotropic metastasis of cancer cells to specific distant target organs [76]. Exosomes involved in modulating the microenvironment of recipient cells via Src phosphorylation enhances the expression of S100 to promote metastasis [72,106,107]. Astrocytoma metastasis is mediated by the exosomal miRNA as it could promote microenvironment-induced loss of PTEN expression in brain tissues [72]. A plethora of scientific evidence elucidated the tumor cell-derived exosomes in influencing “stromal cells, VECs, and fibroblasts’ to aid uncontrolled tumor growth. For instance, a few reports by Annette Becker et al. (2016) described the role of exosomes in enhancing metastasis of cancer by modulating vascular permeability [108]. Exosomes loaded with miR-105 derived from metastatic breast cancer cells could impair the endotheliocyte expression of ZO-1 protein consequently resulted in the enhanced sensitivity to cancer [109]. This kind of exosomal activity can be observed even in hepatic metastasis of breast cancer [109,110]. PDAC-derived exosomes are characterized by the higher expression of MIF to promote suitable microenvironment formation by modulating to TGF-β and fibronectin levels for enhancing the metastasis and growth of premetastatic niche [111,112]. Tumor-derived exosomes (TDEs) regulate lymph node metastasis and melanoma tumor growth. In addition, TDEs also release Rab3D, TGF-β1, and LMP1 factors to enhance EMT and consequently promote the invasion of tumor cells [113,114,115].

## 5. Exosomes and Cancer Stem Cells (CSCs) in Invasion and Metastasis

Stem cell-derived exosomes (SDEs) play a significant role in the progression of cancer through invasion and metastasis. SDEs can aid paracrine action via exchange of genetic components, thereby playing a significant role in tumor cell metastasis [116,117]. CSCs-derived exosomes are characterized by the presence of immunogenic function due to Rab GTPases, several signal transduction components, and annexins, whereas the MSCs-derived exosomes are characterized by their non-immunogenic nature [118]. However, both stem cell-derived exosomes could influence the induction of tumor pre-metastatic niche, tumor metastasis, and tumor growth. Several other scientific evidences also described the significance of MSCs in conferring stem cell signatures and EMT to augment the cancer cell survival [119]. A scientific report by Wang M, Zhao C et al. (2014) elucidated the role of MSC-derived exosomes by demonstrating that they could deliver miR-221 to “metastatic lymph node of GC cell line” HGC-27 and confer an uncontrolled proliferation property followed by the metastasis in gastric cancers [120]. In the case of MCF-7 breast cancer cells, the MSC-derived exosomes foster the modulation of Wnt-signaling to confer metastasis and continuous cancer cell division [121]. miRNA-140 has a vital role in regulating tumor growth and metastasis [122]. Downregulation of miRNA-140 levels confer a substantial rise in CSCs and metastasis of breast cancer cells via blockade of the tumor suppressive pathways [122]. On the contrary, the exosomes released from ductal carcinoma in situ (DCIS) were composed of lower levels of miRNA-140 than the whole DCIS cell population, which consequently resulted in the tumor metastasis [122]. Other DCIS exosomal-miRNAs, viz. miRNA-29a and miRNA-21, could mediate the metastasis of cancers [123].

Several reports elucidated the efficacy of exosomes in inducing metabolic reprogramming via the restoration of cancer cell respiration [112]. In the case of lung cancer, the exosomes are reported to promote EMT followed by the metastasis and invasion to colonize distant organs [124]. Another report by Ono M, Kosaka N et al. (2014) described the role of exo-miR derived from bone marrow (BM)-MSCs for inducing dormancy in metastatic breast cancer cell niche by impairing the *MARCKS* gene [125]. MSC-derived exo-miR-143 could mitigate the metastasis of osteosarcoma cells, and all these reports conclude that this area of research could be a promising approach to target the cancer stem cells involved in metastasis [126,127]. Yuanyuan Che et al. (2019) have recently reported the role of exo-miR-143 derived from human BM-derived MSCs in mediating prostate cancer invasion and metastasis by modulating TFF3 [128]. Another report by Dong-Mei Wu et al. (2019) elucidated the role of exo-miR-126-3p derived from BM-MSCs in developing pancreatic carcinoma via the modulation of “*disintegrin and a metalloproteinase-9 (ADAM9)*” [129].

Exosomes derived from MSCs express “*CD63, CD9 and CD81*” to elicit TRAIL-mediated apoptosis in cancer cells in a dose-dependent fashion without cytotoxicity to human bronchial epithelial cells [130,131]. Another report by Lou G et al. (2015) showed that “exo-miR-122 from adipose tissue-derived MSCs” can confer HCC cell sensitization to sorafenib by modulating gene expression [132]. Exosomal siRNA derived from MSCs acts against the sorafenib-resistant CSCs of HCC by impairing GRP78 [133]. Exo-miRNA-119a derived from MSCs could impair the invasion and metastasis of glioma cells by reducing the expression of ankyrin repeat and PH domain 2 [134]. A report by Lang FM et al. (2018) demonstrated the natural ability of MSCs in producing exosomes with miRNA-124a to act against glioma stem cell lines [135]. MSCs can also generate the miR-16-5p loaded in exosomes to impair the proliferation, invasion, and metastasis of colon cancer cells [136]. MSCs derived exo-miR-101-3p could inhibit the proliferation, invasion, and migration of oral cancer cells by targeting the collagen type-Xα1 chain [137].

## 6. Exosomes and Cancer Cell Survival

Cancer cells generate copious levels of exosomes; the Munc13-4 is a Ca^2+^-dependent regulator of Rab-dependent signaling, which confers increased CD63+, CD9+, and ALIX+ exosomal release in cancer cells, which enhances their survival [138]. Exosomes derived from cancer cells are characterized by the expression of RTKs, EGFR, and HER-2. Knockout of EGFR or/and HER-1 can impair the exosomes-mediated MAPK cell survival signaling in “tumor-associated monocytes” as they promote cancer cell survival [139]. The results of this study demonstrated that cancer cell-derived exosomes activate MAPK signaling in tumor-associated monocytes via the transport of active RTKs to block the activities of caspases [139]. Hence, targeting cancer cell-derived exosomes is a novel strategy to develop novel anticancer therapeutic interventions for blocking cancer cell survival.

## 7. Exosomes in the Process of Angiogenesis

Exosomes derived from malignant mesothelioma possess oncogenic cargo and are reported to be involved in angiogenesis of cancer cells by enhancing the blood vessel regeneration, movement of VECs and fibroblasts [140]. A report by Cui H et al. (2015) has described the role of exosomes derived from lung adenocarcinoma to mediate angiogenesis by modulating the expression of *Ephrin α3* via miR-210-dependent fashion [141]. The authors of this study demonstrated that overexpression of TIMP-1 in tumor cells enhanced the accumulation of exo-miR-210 in a “CD63/PI3K/AKT/HIF-1-dependent signaling” and aid in the tube formation ability in HUVECs, which consequently augmented neovascularization in “A549L-derived tumor xenografts” [141]. Exosomes are composed of angiogenic factors for efficient vascular endothelial migration, proliferation, and formation of basement membranes, which promotes the synthesis of neovascularization networks towards tumor cells during nutrient and oxygen deprivation. For instance, MSC-derived exosomes enhance angiogenesis towards tumor cells by promoting the activation of ERK1/2 and p38-MAP Kinase signaling [142]. Prior reports have demonstrated the extensive activity of HIF-1α during hypoxia to release high exo-miR-210 from metastatic cancer cells for angiogenesis [28,143,144].

Another report by Salomon C et al. (2013) reported the role of exosomes derived from placental MSCs in vasculogenesis and angiogenesis based on the oxygen tension [144]. Tatiana Lopatina et al. (2014) described the role of EVs derived from adipose mesenchymal stem cells (AD-MSCs) in angiogenesis. PDGF is another factor which could enhance the release of EVs to mediate angiogenesis [145]. Exosomes derived from “*MSC marker CD105-positive cells*” of human renal carcinoma can aid angiogenesis by stimulating endothelial growth and vascular networks, and foster the formation of premetastatic niche [117]. This was confirmed by the in vivo implantation of CD105-positive cancer stem cells into SCID mice [117]. However, certain conflicting reports raised queries pertaining to the role and mechanism of exosomes in angiogenesis. For instance, a study reported that the exosomes derived from MSCs could impair angiogenesis by downregulating VEGF expression in breast cancer cells [146,147]. In addition, the MSC-derived “exo-miR-16” was reported to mitigate VEGF expression in 4T1 cells and modulate angiogenesis [148,149,150]. Hence, the prospective anti-cancer therapeutic modalities should focus on targeting the above exosomal signaling to mitigate angiogenesis.

## 8. Exosomes and Radio- and Chemo-Resistance

### 8.1. Mesenchymal Stem Cells-Derived Exosomes and Chemoresistance

Development of drug resistance occurs when CSCs are exposed to the repeated administration of drugs/radiation [133]. Since exosomes can carry small molecules such as siRNAs, miRNAs, lncRNAs, and genetic materials, they can be used as specific vehicles to target signaling pathways, viz. “*Wnt, Notch, Hippo, and Hedgehog*”, responsible for CSCs signatures [151].

Exosomes derived from tumor microenvironment also modulate EMT, and chemoresistance [152,153]. Chemotherapy can elevate exosomal secretions in cancer cells, resulting in the transfer of chemoresistance-related miRNA (*miR-100, miR-222, miR-30a, miR-17*) and mRNA to the adjacent cancer cells to induce susceptibility to chemotherapy. On the other hand, the MSC-exosomes could also mediate the drug efflux and transfer of chemoresistance property to neighboring cells through “MRP2, ATP7A, and ATP7B” [3,154,155]. For instance, the exosomes derived from MSCs mediate the chemoresistance in gastric cancer cells to 5-FU by enhancing the expression of multidrug resistant proteins, viz. MDR, MRP, and LRP, as well as by activating CaM-Ks/Raf/MEK/ERK signaling [11]. The exosomes derived from MSCs were shown to impair the sensitivity of BM2 cells to docetaxel [125]. MSC-exosomes loaded with synthetic “anti-miR-9” can reverse the temozolomide-induced chemoresistance in GBM cells [125]. Another report by Lou, G et al. (2015) elucidated the efficacy of ADMSC-exosomes loaded with miR-122 for inducing susceptibility of HCC cells to chemotherapeutic agents, 5-FU or sorafenib; this is a novel approach to enhance the sensitivity of HCC cells to chemotherapy [132]. In a separate study, it was reported that MSC-exosomes obtained from AML patient samples had high levels of “*miR155 and miR375*”, which could confer chemoresistance upon the repeated administration of cytarabine and AC220 (an FLT3 inhibitor) [156,157]. Another report by Yu Zhou et al. (2019) demonstrated the role of BM-MSC-derived exosomes in delivering gemcitabine and paclitaxel into chemoresistant PADC cells [158]. Exosomes derived from gemcitabine (GEM)-resistant CSCs of pancreatic carcinoma enhance the chemoresistant traits in GEM-sensitive pancreatic cells through the delivery of miR-210 [159] (Table 1).

### 8.2. Astrocyte-Derived Exosomes and Radioresistance

Astroctye-derived exosomes (ADEs) are the key exosomes involved in sending/receiving signal communication inside the nervous system [165]. However, to date, there are no reports highlighting the role of ADEs in mediating chemoresistance in brain cancer. Certain research reports described the role of miRNA-26a in ADEs to regulate several neurological diseases, including gliomas, in tumor microenvironments, by enhancing de novo tumor formation and radio-sensitivity through the “suppressor of PTEN and ATM” [165,166,167]. ADEs with miR-19a can induce the downregulation of cancer cell PTEN-expression consequently enhance the CCL2 secretion and myeloid cell recruitment to facilitate metastasis of brain cancer cells [72]. Normal astrocytes inside the tumor microenvironment can mediate tumor cell proliferation and growth by transferring exosomes containing alpha-crystallin B chain (CRYAB). The ADEs derived from astrocytoma cells could transfer CRYAB to promote the chemoresistance of tumor cells [168].

### 8.3. Macrophage-Derived Exosomes and Chemoresistance

Tumor-derived exosomes are significantly involved in mediating chemoresistance of cancer cells to several chemotherapeutic agents and enhance stem cell signatures by modulating several cell signaling pathways and the expression of genes [169,170,171]. TAMs-derived miR-223 has a significant role in the acquisition of drug resistance, since these exosomes possess a substantially higher level of miR-223 [172,173]. A report by Xiaolan Zhu et al. (2019) elucidated the role of TAMs-derived exosomes during hypoxia to promote chemoresistance of EOCs (epithelial ovarian cancer cell) to chemotherapeutic agents by transferring miR-223. Mechanistically, hypoxic EOC cells conferred the M2-polarization of macrophages and enhanced the expression of exo-miR-223 in TAMs, which modulated the “PTEN/PI3K/AKT” proteins that are involved in the chemoresistance of EOC cells [160,161,162]. The understanding of TAMs-derived exosomal responses to chemotherapy drugs is a novel approach to develop chemoresistant CSCs. TAMs-derived exosomes foster cisplatin resistance in gastric tumor cells by transferring miR-21 to regulate PTEN/PI3K/AKT cascade. Stromal cell derived exo-miRNA-21 could be transferred to EOCs in omental tumor microenvironments and facilitate the chemoresistance against chemotherapy [163] (Table 1).

### 8.4. Myeloid Leukemia Cell-Derived Exosomes and Chemoresistance

AML (acute myeloid leukemia)-derived exosomes foster the chemoresistance of AML cells to cytarabine, as they secrete VEGF/VEGFR factors to promote glycolysis in human umbilical vein endothelial cells (HUVECs) [174]. Several reports described the EV-mediated resistance transfer from resistant AMLs to sensitive AMLs by miRNAs for modulating expression of apoptotic proteins [175,176,177]. AML-exosomes can actuate bone marrow stromal cells to secrete IL-8, that can further provoke AML cells to acquire chemoresistance to etoposide [178,179,180]. EVs derived from promyelocytic leukemia cells are reported to be involved in the transfer of chemoresistance properties from resistant strains to sensitive strains due to the direct transfer of multidrug resistance protein 1 (MRP-1) and miRNAs [177]. The transfer of Exo-miR-365 derived from drug-resistant Chronic Myeloid Leukemia (CML) cells to the CML-sensitive cells resulted in the acquisition of imatinib chemoresistance in sensitive cells (Table 1) [181,182].

### 8.5. Cancer-Associated Fibroblast-Derived Exosomes and Chemoresistance

Cancer-associated fibroblasts (CAFs) play a vital role in inducing chemoresistance to tumor cells [183]. For example, the chemoresistance is acquired upon repeated treatment of gemcitabine to pancreatic cancer cells via pancreatic CAFs-derived exo-miR-146a and Snail [129,184,185]. In addition, the transfer of “CAFs-derived exo-miR-21” to cancer cells via exosomes is reported to promote chemoresistance, as these nucleic acids activate PI3K/AKT signaling and APAF-1 pathway [186,187]. CAFs can also promote the chemoresistance and expression of stem cell signatures in colorectal cancer cells by enhancing the activity of CAFs-derived exosomal Wnts [188] (Table 1).

## 9. Exosomes and Chemoresistant-Cancer Therapeutics

Exosomes can carry information by delivering their components to the adjacent cells and modulate the bio-functional reprogramming of neighboring cells [151]. Exosomes play a prominent role in mediating intercellular communication to enhance chemoresistance in several tumors, including glioblastoma [189], acute/chronic leukemia [190], lung cancer [191], breast cancer [192], PDAC [193], EOC [187], and prostate cancer [194]. Exosomes confer chemoresistance and stemness by delivering drug resistant MDR-1, P-gP [195], survivin [196], and UCTH-L1 (ubiquitin carboxyl terminal hydrolase-L1) [192] to the neighboring cells [197].

Chemoresistance induces cancer stemness through exosomal secretions like peptides, nucleic acids, and other small molecules in several cancers and consequently makes the drug-sensitive cells into CSCs [197]. The exosomes derived from CSCs further transfer their molecular components to non-CSCs to maintain CSCs dynamic equilibrium within the tumor microenvironment [198]. In addition, various factors derived from MSC-exosomes enhance stemness to chemotherapy by modulating several signaling pathways in cancer cells [11,199]. Hence, targeting exosomes might be an effective approach to develop chemoresistant-cancer therapeutics against both CSCs and non-CSCs to break this dynamic equilibrium [199]. Exosomes derived from prostate cancer cells can induce fibroblast proliferation, angiogenesis, and tumor growth [200]. Exo-miR-155 derived from PDACs can convert normal fibroblasts into CAFs [201]. A report by Donnarumma et al. (2017) described the efficacy of exo-miRNAs (miR-21, miR-378e and miR-143) derived from CAFs for enhancing the stemness signatures and EMT of breast cancer cells [198]. Another report by Boelens MC et al. (2014) elucidated the role of stromal cell-derived exosomes to enhance the chemoresistance by activating STAT-1 signaling and Notch-3 signaling in breast cancer cells [202]. In addition, the exosomes derived from CRC-initiating cells can mediate the transfer of *claudin-7* to neighboring cancer cells and further enhance the invasion and metastasis [203]. Exosomal HIF-1α derived from nasopharyngeal cancer cells can enhance the metastasis and invasion [204]. CLIC1 was highly expressed in exosomes derived from CSCs to enhance the GBM cell division and growth [205].

The stemness of GBM cells is promoted by the exo-miR21 [204,206]. Exo-miR-200 derived from breast cancer cells significantly enhances the stemness, EMT of adjacent cells [98]. Exo-miR-21 and Exo-miR-155 exert a significant role in the cross-talk between neuroblastoma cells and human monocytes to actuate chemoresistance via “exo-miR-21/TLR8-NF-κB/exo-miR-155/TERF” signaling cascade [207]. Exo-long non-coding RNA (lncRNA) derived from cancer cells involved in the cancer cell proliferation, progression, and angiogenesis. Furthermore, the blockade of nSMase activity using RNA interference methods could mitigate exosome production and prion delivery to reduce metastatic colony formation. Knockdown of the underlying factors for ESCRT machinery is a beneficial strategy to regulate exosomes biogenesis in cancer cells [208,209]. Furthermore, the exosomes encapsulated with therapeutic molecules can effectively target chemoresistant CSCs by modulating the signaling pathways responsible for stemness, viz., “Wnt, Notch, Hippo, Hedgehog, NF-κB, and TGF-β pathways” [210,211,212,213].

Exosomes are efficient nanometric vehicles to carry small molecules as therapeutic interventions against several diseases including cancers [20]. They have theranostic applications since they are nonimmunogenic and possess robust nano-delivery capability and can be engineered to carry small molecule therapeutics like nucleic acids, peptides, antibodies, and proteins against CSCs, and multiple diseases [7,20]. For instance, tumor antigens, apoptosis-promoting proteins [53,213], mutant proteins related to apoptosis are transferred through exosomes as nanobodies into the cancer cells [214]. In addition, transferrins, immuno-proteosomes, and lactoferrins can be delivered as small molecule therapeutics against several cancer cells [215,216,217]. Dendritic cells (DCs) are engineered to possess the enhanced expression of fusion proteins like “αv integrin-specific iRGD peptide and Lamp2b”. Exosomes derived from these cells exhibit a higher surface expression of iRDG [217]. The engineered DCs with the above exosomes conferred a significant chemotherapeutic drug delivery and produced anti-breast cancer efficacy [217]. A report by Luketic et al. 2007 described the efficacy of exosomes derived from peptide-pulsed DCs for enhancing the T-cell immune activity by presenting antigens [218]. Aspe et al. (2010) elucidated the role of exosomes loaded with survivin-T34A for promoting apoptosis in PDACs and induce cancer cell susceptibility to gemcitabine [214,219].

Exosomes enriched with several miRNAs can modulate cancer cell/CSCs survival, invasion, and metastasis; and the development of therapeutic molecules to intervene with these exosomes can enhance the sensitivity of several cancer cells/CSCs to chemotherapeutic drugs [199,220]. Exosomes that can deliver siRNAs to silence the genes in cancer cells have been explored by several research reports. For instance, the exosomal delivery of siRNA blocked RAD51 consequently impaired breast cancer cell proliferation [221]. Likewise, siRNA delivery through exosomes silences PLK-1 and mitigates proliferation of bladder cancer [222]. Similarly, siRNA transfer through exosomes could silence c-Myc and promote pro-apoptotic protein activation in lymphoma cells [223].

## 10. Combination Strategies Testing Phytochemicals with Exosomes for the Treatment of Cancers

Several phytochemical therapeutic cargos can be loaded into exosomes for cancer therapy. For example, the anticancer drug paclitaxel (PTX), which is derived from *Taxus brevifolia* and exosomes-loaded PTX could inhibit drug-resistant CSCs with much higher efficacy (>50-fold) compared to free drugs [224]. Exosomes-loaded PTX exert its efficacy in mitigating lung carcinoma by declining pulmonary metastasis [224,225]. In addition, the “exosomes-encapsulated PTX” induces cell death to autologous cancer cells. Interestingly, the exosomal PTX-pretreated donor cells could produce PTX-encapsulated exosomes to enhance cytotoxicity [226]. Munagala et al. 2016 have described the efficacy of “exosomes-encapsulated Withaferin A” to mitigate cancer cell proliferation and angiogenesis in human lung cancers [227]. Celatrol is a triterpenoid isolated from plant, *Tripterygium wilfordii*. “Exosomes-encapsulated celastrol” could mitigate the tumor cell progression in human lung cancer xenograft models than free celastrol groups [228,229]. Curcumin, a widely used anti-cancer and anti-inflammatory agent, has solubility limitations due to its hydrophobic nature [230]. “Exosomes-encapsulated curcumin” could easily incorporate curcumin to cross lipid layers of cancer cell membrane to promote anti-tumor effect [230]. A report by Zhang HG et al. (2007) reported the efficacy of “exosomes encapsulated with curcumin” in inducing the impairment of NK cells and reversed the susceptibility of breast cancer cells to several chemotherapeutic drugs [231].

## 11. Dietary Exosomes and Cancers

Several reports have described the dietary molecules such as long-chain polyunsaturated fatty acids (PUFAs), polyphenols, flavanoids, Nelumbo alkaloids, carotenoids, resveratrol, EGCG, vitamins, folates, and curcumin as chemopreventive agents, as they could modulate several “exosomal-miRNAs” involved in mediating several cancers [232,233,234,235]. Recent scientific evidence reported the efficacy of dietary exosomes in mitigating cancer growth. For instance, the “grape exosome like nanobodies” could mitigate DSS-induced colitis by inducing proliferation of intestinal stem cells [236]. In addition, milk-derived exosomes can be significantly employed for loading the curcumin in order to augment the transport of curcumin through the lipid bilayers of tumor cells [237]. However, there are reports describing the progression of HCC upon the consumption of cow’s milk due to the presence of Exosomal miR-21, which can effectively induce IL-6-mediated STAT-3-dependent miR-21 transcription [238,239]. Milk derived exo-miR-155 can enhance STAT-3-dependent tumor growth [240]. A report by Zhuang X et al. (2011) demonstrated the efficacy of “exo-cur” to attenuate brain inflammatory diseases, and its efficacy was proven in an LPS-induced brain inflammation model, autoimmune encephalitis models, and a GL26 brain tumor model when administered with the “exo-cur” intranasally [241]. This formulation induced the rapid delivery of curcumin into the brain and promoted the apoptosis of microglial cells to mitigate inflammation [241].

## 12. Exosomes and Oxidative Stress-Mediated AD, PD, and Stroke

Oxidative stress can enhance the release of exosomes from multivesicular bodies (MVBs) in several neurodegenerative and demyelinating diseases (multiple sclerosis) [242,243], cerebral ischemia [244,245], and brain tumors [246]. A plethora of research reports elucidated that the oxidative stress is associated with pathogenesis of neurodegenerative diseases, viz., AD, PD, etc. A significant characteristic of AD is the continuous occurrence of chronic neuroinflammation, oxidative stress followed by the production of Aβ plaques and neurofibrillary tangles [247,248,249]. These events actuate the irreversible dysfunction of neurons. Furthermore, the oxidative stress could confer the enhanced production of Aβ in AD through the high exosomal release of β- and γ-secretase as a result of the substantial rise in amyloid precursor protein (APP) metabolism further leading to generation of β-carboxyl-terminal fragments (CTFs) [250,251,252]. Exosomes also induce the aggregation of Aβ and plaque formation. A recent review by Zhi-You Cai et al. (2018) elucidated the role of oxidative stress in enhancing both beta-amyloid pathogenesis and the hyperphosphorylation of tau proteins. In addition, they reported the ability of exosomes in clearing beta-amyloid [242] (Figure 2). Hence, the therapeutic modalities targeting exosomes could deliver promising clinical outcomes for AD.

Miranda et al. (2018) described the impairment of autophagy and lysosomal activity with neuronal Vps34 disruption, which simultaneously resulted in the release of unique exosomes with beta-amyloid, APP, and β- and γ-secretases [253,254]. The propagation of oxidative stress-related AD also pertains to the delivery of ubiquitin ligases, APP-CTFs, amyloid proteins, and secretases through exosomes derived from damaged neurons [255,256,257]. Exosomes enriched with Alix, flotilin-1, hyperphosphorylated tau extensively observed in the plaques obtained from AD patients; indicating that exosomes are considered as the nucleation centers for Aβ formation [258,259]. Exosomes composed of PrP^c^ could mediate the beta-amyloid fibrillation followed by synaptotoxicity [260]. Furthermore, astrocyte-derived exosomes of AD patients were composed of neurotoxic cargo, i.e., “β-secretase/γ-secretase” and “sAPPβ”, upto 20 times higher than neuron-derived exosomes [261]. MSC-derived exosomes encapsulating with small molecule therapeutics can effectively target beta-amyloid, since AD-MSC-derived exosomal neprilysin can degrade beta-amyloid [262].

Mitochondria-mediated oxidative stress has significant implications in the underlying mechanisms of α-syn aggregation during PD [263,264]. In addition, the miRNAs in CSF-derived exosomes can induce significant alterations in the KEGG pathway, as well as in dopaminergic and cholinergic synapses in PD patients [265]. Exosomes deliver toxic misfolded α-syn in between dopaminergic neurons and induce apoptosis, consequently causing PD pathogenesis. Exo-miR-137 derived from serum could affect oxidative stress inside the neurons through OXR1 during PD [266] (Figure 2). However, there are certain reports where exosomes also possess neuroprotective effects in PD [267].

Exosomes exert significant implications during ischemic and hemorrhagic strokes (Figure 2). For instance, the BM-MSC-derived exo-miRNA-138-5p fosters a neuroprotective effect to astrocytes during ischemic stroke by modulating the role of LCN2 [268]. MSC-derived exosomes also promote the recovery of neurogenesis and angiogenesis during stroke [269]. In addition, the miRNA-19 and miRNA-124 derived from exosomes are significant diagnostic markers of stroke, and could be considered as direct alternatives to therapy [270]. Exosomes derived from ischemic astrocytes treated with ‘semaphorin 3A Inhibitor’ potentially augment the stroke recovery by enhancing the axonal outgrowth and the expression of prostaglandin D2 synthase [271]. Furthermore, the levels of exosomal miR-223, miR-21-5p, miR-9, miR-124, miR-30-5p, and miR-134 are reported to exhibit a potential role in stroke pathophysiology and post-stroke complications [272]. Hence, these exosomal cargos can be implicated as diagnostic/prognostic markers in stroke pathophysiology [272].

## 13. Exosomes and Targeted Drug Delivery

Although exosomes contain DNA and RNAs to deliver into the target cells for genetic modifications during biological/pathogenic processes, the exosomes can also be used as the delivery vehicles for nucleic acids [273]. These are the significant features of exosomes which are beneficial aspects for gene therapy in treating cancers and oxidative stress-related neurodegenerative diseases [273]. Exosomes can be used as therapeutic vehicles to carry exogenous genetic components, such as miRNAs, and siRNA to damage/knockdown genes of interest in gene therapy against cancers, neurological diseases, etc. [91]. Exosomes could exhibit CSC-specific antigen proteins for T cell activation as well as for anti-CSC immunization [60]. In addition, these are natural nanovesicles used for carrying exogenous genetic components to target several CSC-specific cell signaling pathways, viz. Wnt pathway, Notch, Hippo, Hedgehog, etc. [151]. Hence, it is recommended to carry out further research investigations for developing exosomes to target CSC-specific signaling pathways.

The modification of the exosomes is suggested by using surface display technology to display candidate proteins or particular surface receptors for efficient cell recognition, as well as for specific targeting [274]. For instance, the exosomes with Tspan8 could efficiently bind to CD11b and CD54 positive cells [32]. Engineering of the exosomes with donor cells was performed by several researchers to express cell specific proteins or peptides fused with exosomal membrane proteins such as “Lamp2b, CD-9 and tetraspanins CD-63”, which further place the candidate proteins on the surface of exosomes [274]. Another report by Alvarez-Erviti et al. 2011 described the role of exosomes in carrying “neuron-specific rabies viral glycoprotein (RVG) peptide”, which is able to knockdown certain genes in neurons and glial cells by the selective delivery of siRNA; this peptide can also bind to the Ach receptors on neuronal cells. Hence, engineered exosomes are selective tools in treating neuronal cancer [91]. Recent research reports described the role of engineered magnetic exosomes in combating tumor growth [275]. Engineered anti-EGFR nanobodies with GPI exosomal fusion proteins are transfected into donor cells to target (+) EGFR-tumor cells [276]. Another report by Watson et al. 2016 described the substantial rise in the accumulation of drug-loaded exosomes in tumor cells by the blockade of a monocyte/macrophage uptake receptor, which normally mediates drug scavenging in liver cells [277].

Furthermore, exosomes are preferred nanometric vehicles for establishing nanodrug delivery systems (NDDS) [278]. Due to their long lasting half-life, and ability in targeting tissues, biocompatibility, and limited toxicity, exosomes are the preferred choice materials to carry and deliver chemotherapeutic drugs, antimicrobial agents, analgesics, neurological drugs, etc. [279,280]. For instance, exosomes have been tested to deliver phytochemicals such as curcumin (a polyphenolic compound) [281]. Curcumin is a potent anti-oxidant, chemopreventive, and anti-inflammatory agent [281,282,283], but suffers from poor solubility and cellular uptake [281]. The efficacy of curcumin has been increased by preparing a nano structured “Exo-cur”—an exosomal curcumin [284,285]. Exo-cur induced a significant decline in the inflammatory cytokines, viz. IL-6, and TNF-α, compared to the macrophages treated with curcumin alone [286]. Thus, the ability of exosomes in carrying hydrophobic molecules such as curcumin has been explored to enhance the anti-inflammatory activity [282]. The intricacy of the exosomes can be determined by labeling exosomal protein markers—viz. TSG101 and CD81—on the exosome-curcumin complex [286]. Celastrol, taxol derivatives, and doxorubicin are small molecules characterized by a low solubility and short half-life which are leading to the poor therapeutic efficacy of drugs in targeting cancers [287,288]. Recent research reports on zebrafish models demonstrated that the encapsulation of chemotherapeutic drugs such as paclitaxel and doxorubicin into exosomal complexes isolated from various cell lines (“gliobastoma astrocytoma U-87 MG, endothelial bEND.3, neuroectodermal tumor PFSK-1, and glioblastoma A-172”) could effectively deliver these small molecules into the brain and blood–brain barrier to treat brain cancers [273,289,290]. In conclusion, the efficacy of otherwise poorly soluble drugs can be improved by exosomes.

Exosomes can target cancer cells with ten-fold higher efficacy than liposomes due to better ligand–receptor interactions at the cancer cells [287,288]. Several reports have demonstrated that the engineering of exosomes in vitro with specific ligands to target cancer cells is a potentially viable strategy to improve selectivity and reduce the systemic toxicity of drugs [288]. Tian et al. (2014) described the efficacy of exosomes with “αv integrin-specific iRGD peptide” for enhancing the anti-cancer effects of doxorubicin in vivo than the free drug group [289]. Exosomes are suitable vehicles to carry chemotherapeutic molecules to target chemoresistant CSCs [291]. Hence, it is essential to determine the efficacy of exosomes with novel chemotherapeutic molecules to circumvent tumor MDR, angiogenesis, and metastasis through targeted delivery. Another study described the efficacy of paclitaxel encapsulated into macrophage-released exosomes in targeting tumor MDR and pulmonary metastasis, which could be attributed to the specific proteins located on the surface of exosomes [224]. The ability of exosomes in overcoming MDR could be due to their endocytosis to bypass the P-gP excretion.

Exosomes can also act as the drug delivery systems for proteins such as catalase [273]. For instance, Parkinson’s disease (PD) is characterized by ROS-mediated oxidative stress, inflammation across the brain, poor antioxidant defense, and microglia activation [291,292,293]. Patients with PD exhibit poor antioxidant enzymes, such as catalase and superoxide dismutase, since these enzymes could mitigate oxidative-related neurodegeneration during PD [294,295,296]. Catalase delivery across BBB is associated with several obstacles, but the exosome-loaded delivery can be an effective strategy for PD therapy [273]. Catalase-loaded exosomes follow a sustained delivery with prolonged circulation time for treating Parkinson’s disease [297]. Another study described the role of exosomes loaded with dopamine to release into the nigrostraital system (substantia nigra) for Parkinson’s disease treatment [297]. Exosomal dopamine enhanced the efficacy 15-fold in relation to the free drug group [297]. In addition, the blood exosomes exhibit “transferrin receptor” (TfR) to enhance the drug accumulation, which may be attributed to the natural ability of blood exosomes in targeting neurological diseases without any modifications [297].

Since the blood–brain barrier and brain cells such as astrocytes and pericytes can induce P-gP-mediated drug efflux out of the target cells, thereby enhancing the pathogenesis of brain diseases, the use of exosomes assists in facilitating the excellent penetration of CNS drugs towards target cells and overcome the problem of drug efflux [254,298,299,300]. Macrophage-derived exosomes (MDEs) were reported to interact with “ICAM-1 & the carbohydrate-binding C-type lectin receptors (CBCLR)” on the brain microvessel endothelial cells and attribute to the uptake of MDEs to target inflammatory cells [300]. Similarly, the macrophage-derived exosomes loaded with BDNF can be conducive to the alleviation of neuroinflammation during neuroinflammatory diseases [300]. Thus, exosomes are the preferred choice of vehicles to treat neurodegenerative diseases, cerebrovascular diseases, and brain tumors [297].

## 14. Conclusions

Exosomes play a major role in modulating cancer cell growth, invasion, metastasis, and the development of stemness signatures on chemoresistant cancer cells. Several phytochemicals and small molecule therapeutics encapsulated into exosomes can effectively reach target cells and deliver effective therapeutic outcomes in patients suffering from multiple diseases, such as neurological diseases, cancers etc. Furthermore, better understanding of the involvement of specific exosomal cargo in chemoresistance and CNS neurodegenerative diseases is a significant aspect enabling enhanced designs of novel molecular therapies, biomarker discovery for early diagnosis, and treatment optimization.

## Figures and Tables

**Figure 1 ijms-21-06818-f001:**
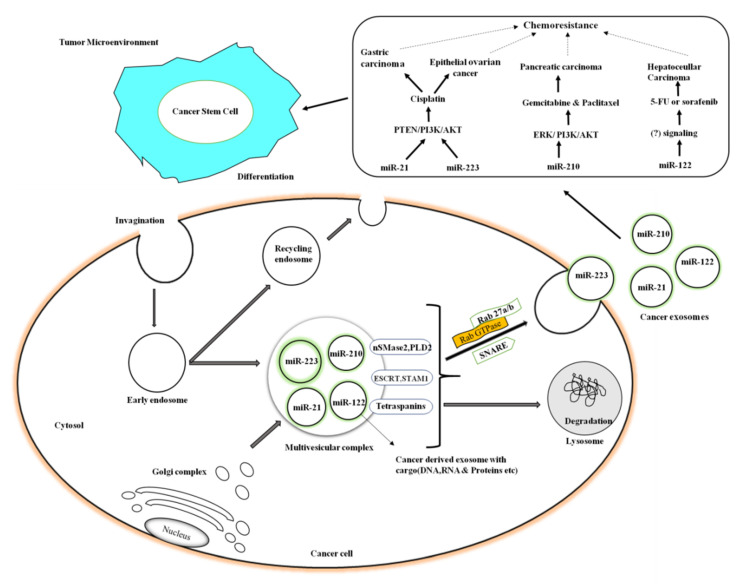
Schematic representation of cancer cell exosomes generation with cargo mediated through endosomal sorting complexes: The exosome-enriched miRNAs confer the chemoresistance (stemness) within the tumor microenvironment in multiple cancers viz., gastric cancer, epithelial ovarian cancer, pancreatic carcinoma, and lung carcinomas.

**Figure 2 ijms-21-06818-f002:**
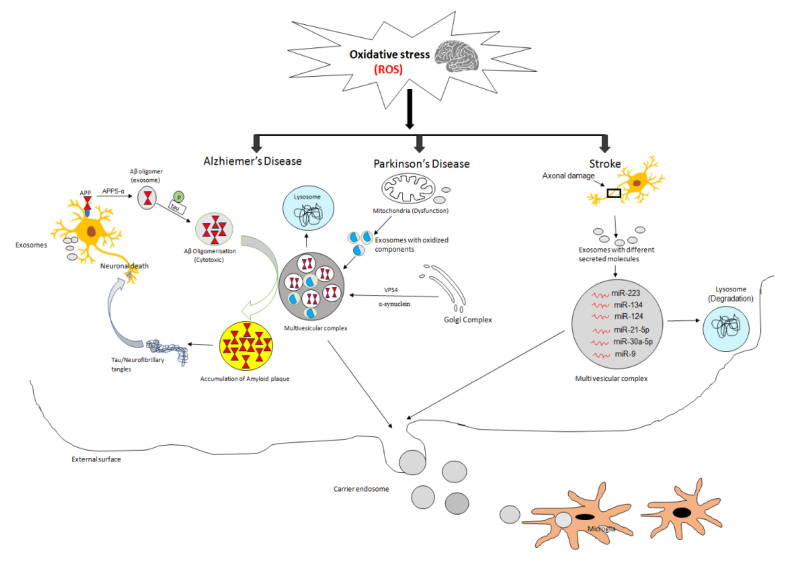
Schematic representation of the exosomes-mediated pathophysiology of neurodegenerative diseases ROS-mediated oxidative stress could foster the generation of amyloid precursor protein (APP) due to the extensive exosomal release of β-secretase/γ-secretase during AD pathophysiology. Exosomes-mediated pathophysiology of PD is accompanied by the accumulated toxic misfolded α-syn in between dopaminergic neurons and foster apoptotic events. Exosomal cargo with different secreted miRNAs is associated with ischemic stroke and could be used as a novel diagnostic and prognostic marker for stroke.

**Table 1 ijms-21-06818-t001:** List of exosomal miRNA enhancing the chemoresistance or stemness in cancers: Upward arrow indicates “upregulation”; downward arrow indicates “downregulation”.

Exosomal miRNA	Chemoresistant Cancer	Chemotherapeutic Drug	Expression Profile	Target Pathways	Ref.
miR-122	Hepatocellular carcinoma	5-FU or sorafenib	-	-	[132]
miR155 and miR375	Acute Myeloid Leukemia	Cytarabine	-	-	[156,157]
miR-210	Pancreatic carcinoma	Gemcitabine and Paclitaxel	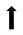	ERK and PI3K/AKT	[158]
miR-223	EOCs (epithelial ovarian cancer cell)	Cisplatin	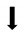	PTEN/PI3K/AKT	[160,161,162]
miR-21	Gastric carcinoma	Cisplatin	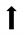	PTEN/PI3K/AKT	[163]
miR-365	CML (Chronic myeloid leukemia)	Imatinib	_	_	[11]
miR-1246, miR-23a etc.	Breast cancer	Docetaxel	_	MAPK, Wnt, TGF-ß pathways	[164]

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
