# Peer review of "The Role of Exosomes in Stemness and Neurodegenerative Diseases—Chemoresistant-Cancer Therapeutics and Phytochemicals"

_ijms, 2020, doi:10.3390/ijms21186818_

Round 1

Reviewer 1 Report

The review work by Beeraka et al. attempt to describe the role of exosomes in stemness and neurodegenerative diseases - chemoresistant-cancer therapeutics and phytochemicals, as put forward in the title. I have numerous comments regarding the structure of this manuscript that significantly decrease enthusiasm:

  • The link between stemness, neurodegenerative diseases, chemoresistant-cancer therapeutics and phytochemicals was not presented in a satisfactory way.

  • The manuscript overlooked key citations from the ISEV community and did not implement important recommendations for extracellular vesicles nomenclature, caution in data interpretation, and acknowledgment of lack of proper exosome isolation method.

  • The abstract is poorly structured.

  • The manuscript misses a much needed Introduction section, in which authors should define and layout their work by including the significance, the motivation, what awaits readers, and what is the take away message.

  • Another major issue and a pet peeve is the poor selection of references especially when cited references do not really support the claimed statements. For instance, line 107, the proper Exocarta citations are:

Keerthikumar, S., Chisanga, D., Ariyaratne, D., Al Saffar, H., Anand, S., Zhao, K., Samuel, M., Pathan, M., Jois, M., Chilamkurti, N., Gangoda, L. and Mathivanan, S. ExoCarta: A web-based compendium of exosomal cargo. Journal of Molecular Biology. 2015. 

Simpson, R.J., Kalra, H. and Mathivanan, S. ExoCarta as a resource for exosomal research Journal of Extracellular Vesicles. 2012. 

Mathivanan, S. Fahner, C.J., Reid, G.E., and Simpson, R.J. ExoCarta 2012: database of exosomal proteins, RNA and lipids. Nucleic Acids Research. 2012. 

Mathivanan, S. and Simpson, R.J. ExoCarta: A compendium of exosomal proteins and RNA. Proteomics. 2009. 21, 4997-5000.

But the authors cited instead:

  1. Sun, W.; Luo, J.-d.; Jiang, H.; Duan, D. D., Tumor exosomes: a double-edged sword in cancer therapy. Acta Pharmacologica Sinica 2018, 39, (4), 534-541.
  2. Gangoda, L.; Liem, M.; Ang, C. S.; Keerthikumar, S.; Adda, C. G.; Parker, B. S.; Mathivanan, S., Proteomic profiling of exosomes secreted by breast cancer cells with varying metastatic potential. Proteomics 2017, 17, (23-24), 1600370.
  3. Keerthikumar, S.; Gangoda, L.; Liem, M.; Fonseka, P.; Atukorala, I.; Ozcitti, C.; Mechler, A.; Adda, C. G.; Ang, C.-S.; Mathivanan, S., Proteogenomic analysis reveals exosomes are more oncogenic than ectosomes. Oncotarget 2015, 6, (17), 15375.
  4. Keerthikumar, S.; Chisanga, D.; Ariyaratne, D.; Al Saffar, H.; Anand, S.; Zhao, K.; Samuel, M.; Pathan, M.; Jois, M.; Chilamkurti, N., ExoCarta: a web-based compendium of exosomal cargo. Journal of molecular biology 2016, 428, (4), 688-692.
  5. Cheng, L.; Sharples, R. A.; Scicluna, B. J.; Hill, A. F., Exosomes provide 818 a protective and enriched source of miRNA for biomarker profiling compared to intracellular and cell-free blood. Journal of extracellular vesicles 2014, 3, (1), 23743.
  6. Bruschi, M.; Ravera, S.; Santucci, L.; Candiano, G.; Bartolucci, M.; Calzia, D.; Lavarello, C.; Inglese, E.; Petretto, A.; Ghiggeri, G., The human urinary exosome as a potential metabolic effector cargo. Expert review of proteomics 2015, 12, (4), 425-432.

  • Sections are enumerated without a clear and cohesive goal. Grouping sections may be helpful. For instance, sections 11-14 can be grouped under one section: “exosomes and radio- and chemoresistance”

  • Figures are not novel and miss a complete description. Table 1 needs different font and font size.

  • Unclear sentences: for instance line 66: “Exosome production takes place from exosome biogenesis”. “Exosomics” first introduced line 44, but not defined. Does it mean omics (lipidomics, proteomics, transcriptomics, RNASeq, etc.) of exosomes?

  • etc. 

Minor issues:

  • Significant extra space between words: lines33, 63, 75, 79, 100, 113, 121, 129, 168, 169, 186, 192, 437, 600, etc.
  • Lines 31 and 47: comma comes before not after “etc.”
  • Line 48. Extra word: “is”
  • Line 52 cancer biologists and (instead of comma) neurobiologists
  • Line 76: zip code should be one word otherwise the meaning is different.
  • Line 99: replace “new born babies” by “newborns”
  • Line 111. Extra period.
  • etc. 

Author Response

Dear Reviewer 1:

On behalf of my coauthors, please accept my sincere thanks and gratitude for careful perusal and critical review of our manuscript entitled “The Role of Exosomes in Stemness and Neurodegenerative Diseases - Chemoresistant-Cancer Therapeutics & Phytochemicals”. We have revised the manuscript based upon the reviewers’ comments as well as your suggestion. Adequate care has been taken to accommodate each and every suggestion of the reviewers. An itemized, “point-by-point” reply to all the comments is attached separately where we have clearly presented our specific response and additions, deletions and/or modifications that have been made in the revised text, and highlighted.

Point-by-Point Answers to the Comments

Reviewer #1

Major Comments:

Comment-1: The link between stemness, neurodegenerative diseases, chemoresistant-cancer therapeutics and phytochemicals was not presented in a satisfactory way.

Response: Authors thank the reviewer for this suggestion and included the text to interlink the ‘intricate role of exosomes in promoting chemoreistance (stemness) and neurodegenerative diseases.

Comment-2:   The manuscript overlooked key citations from the ISEV community and did not implement important recommendations for extracellular vesicles nomenclature, caution in data interpretation, and acknowledgment of lack of proper exosome isolation method.

Response: Authors addressed these key issues and this review text was minimized to follow the “word count” and described main focus of the article towards stemness, and neurodegeneration, and corresponding chemotherapies, and phytochemicals in relevance to exosomes.   

Comment-3: The abstract is poorly structured.

Response: Authors structured and improved the abstract. 

Comment-4: The manuscript misses a much needed Introduction section, in which authors should define and layout their work by including the significance, the motivation, what awaits readers, and what is the take away message.

Response: Authors described the motivation, and significance of this article in every section.  

Comment-5: Another major issue and a pet peeve is the poor selection of references especially when cited references do not really support the claimed statements. For instance, line 107, the proper Exocarta citations are:

Response: Authors implemented the referencing for Exocarta citations as per the reviewer’s suggestion at corresponding places.

Comment-6: Sections are enumerated without a clear and cohesive goal. Grouping sections may be helpful. For instance, sections 11-14 can be grouped under one section: “exosomes and radio- and chemoresistance”

Response: Authors implemented the side headings as per the reviewer comments for section: “exosomes and radio- and chemoresistance”

Comment-7: Figures are not novel and miss a complete description. Table 1 needs different font and font size.

Response: Authors implemented suggested changes in the Figure-1 and interlinked with chemo resistance as per all the reviewer instructions.   

Comment-8:   Unclear sentences: for instance line 66: “Exosome production takes place from exosome biogenesis”. “Exosomics” first introduced line 44, but not defined. Does it mean omics (lipidomics, proteomics, transcriptomics, RNASeq, etc.) of exosomes? etc.

Response: Authors corrected the sentence in line 66 and removed exosomics word.     

Minor issues:

Comment-1: Significant extra space between words: lines 33, 63, 75, 79, 100, 113, 121, 129, 168, 169, 186, 192, 437, 600, etc.

Response: Authors have removed the space and aligned correctly.  

Comment-2: Lines 31 and 47: comma comes before not after “etc.”

Response: Authors aligned and implemented changes as per the reviewer suggestion correctly.  

Comment-3: Line 48. Extra word: “is”

Response: Authors eliminated “is” and implemented changes as per the reviewer suggestion.  

Comment-4: Line 52 cancer biologists and (instead of comma) neurobiologists

Response: Authors eliminated “is” and implemented changes as per the reviewer suggestion correctly.  

Comment-5: Line 76: zip code should be one word otherwise the meaning is different.

Response: Authors implemented changes as per the reviewer suggestion correctly. Authors corrected this grammar and typo error.

Comment-6: Line 52 cancer biologists and (instead of comma) neurobiologists

Response: Authors corrected this grammar and typo error.

Comment-7: Line 99: replace “new born babies” by “newborns”

Response: Authors corrected this as per reviewer’s suggestion.

Comment-8: Line 111. Extra period.  etc.

Response: Authors corrected this grammar and typo error.

Reviewer 2 Report

In this review article, the authors discuss recent developments in the field of extracellular vesicles (EVs) in the context of neurodegenerative diseases, stemness including cancer stem cells, their role in cancer and resistance to therapies. Finally, authors summarize the applications of EVs as nanodelivery systems for drug delivery and small molecules.

In general, the review covers important aspects of EV biology and is suitable for publication. However, the arrangement of content needs a revision.

Mainly, as a reader I see the review tries to discuss everything out there, and provides the mixed form of information. And not clear why authors want to put neurovegetative diseases with cancer without creating a rationale. Otherwise one dedicated topic would have been easy to follow either role of EVs in neurodegerative diseases or role in cancer. No doubt authors have put great efforts in data collection, and reviewing, there is always room to improve. I have some suggestions for authors they may wish to arrange the review in logical flow of concepts.

(1). Start first heading with introduction by replacing ‘1. Exosomes features and functions’ by ‘1. Introduction’, introducing the exosomes (better to write extracellular vesicles, EVs) and here simply introduce what EVs are. E.g heterogenous population of secreted vesicles, their basic role in cell to cell communication and the type of content they contain and deliver to other cells, cause diseases and can communicate drug resistance messages to other cells and be used as drug delivery system (briefly). And in the last paragraph of introduction create a link between neurovegetative diseases and cancer and resistance to therapies. 

Then next heading for EVs and stemness. Here briefly include the role of EVs in stemness (PMID: 26649044). 

Next EVs in Neurodegeneration.

(2). Move cancer part after neurodegenerative diseases. And move EVs as nanometric vehicles – NDDS, at the end of manuscript i.e before conclusion

(3). The biogenesis part has been reviewed in 10s of hundreds of reviews already. I would suggest authors to use this space for better discussing the main part of the review.

(4). Figure 2: please describe the legends.

(5). It would be worth including an additional schematic figure for the role of EVs in drug resistance related to cancer part of the manuscript.

Other comments:

(6). Section 5: Exosomes & CSCs. Avoid abbreviations in headings, better to write at full. Similarly, section 13: 13. ML-derived. Write ML at full.

(7). While authors mention the role of EVs in Chemoresistance, I would suggest authors to refer the following article (PMID: 29657282).

(8). Similarly, while authors indicate the roles of EVs in Radioresistance, I recommend authors to refer the following article (PMID: 31349735).

Minor typos:

(9). Please build the abstract in a logical flow of concepts in accordance with content in the main body of the manuscript.

(10). In the abstract: we have profoundly discussed… ! authors may wish to remove the word profoundly from the abstract. Now several journals discourage such statements.

(11). AML: write at full, when first time mentioned in the text.

(12). The manuscript needs revision for typos. Few examples, but whole manuscript needs careful proof reading.

- Page 2, introduction: there is extra space between [2] . Exosomes…..

- Line 111: section 2. .Exosomics

- Please have proofread the manuscript for typos in the revised version.

Author Response

Dear Reviewer 2:

On behalf of my coauthors, please accept my sincere thanks and gratitude for careful perusal and critical review of our manuscript entitled “The Role of Exosomes in Stemness and Neurodegenerative Diseases - Chemoresistant-Cancer Therapeutics & Phytochemicals”. We have revised the manuscript based upon the reviewers’ comments as well as your suggestion. Adequate care has been taken to accommodate each and every suggestion of the reviewers. An itemized, “point-by-point” reply to all the comments is attached separately where we have clearly presented our specific response and additions, deletions and/or modifications that have been made in the revised text, and highlighted.

Point-by-Point Answers to the Comments

Comment-1: Start first heading with introduction by replacing ‘1. Exosomes features and functions’ by ‘1. Introduction’, introducing the exosomes (better to write extracellular vesicles, EVs) and here simply introduce what EVs are. E.g heterogenous population of secreted vesicles, their basic role in cell to cell communication and the type of content they contain and deliver to other cells, cause diseases and can communicate drug resistance messages to other cells and be used as drug delivery system (briefly). And in the last paragraph of introduction create a link between neurovegetative diseases and cancer and resistance to therapies.

Then next heading for EVs and stemness. Here briefly include the role of EVs in stemness (PMID: 26649044).

Next EVs in Neurodegeneration.

Response: Authors included the additional text as per the suggestions of reviewer and highlighted in green.

Comment-2: Move cancer part after neurodegenerative diseases. And move EVs as nanometric vehicles – NDDS, at the end of manuscript i.e. before conclusion

Response: Authors rearranged the text throughout the manuscript as per the suggestions of reviewer.

Comment-3: Move cancer part after neurodegenerative diseases. And move EVs as nanometric vehicles – NDDS, at the end of manuscript i.e. before conclusion.

Response: Authors rearranged the text throughout the manuscript as per the suggestions of reviewer.

Comment-4: The biogenesis part has been reviewed in 10s of hundreds of reviews already. I would suggest authors to use this space for better discussing the main part of the review.

Response: Thank you for suggestion. We have included minimal text to delineate the biogenesis of exosomes and we have not expanded further.

Comment-5: Figure 2: please describe the legends.

Response: Authors included the legend for figure-2.

Comment-6: It would be worth including an additional schematic figure for the role of EVs in drug resistance related to cancer part of the manuscript.

Response: Authors included extra figure to depict chemoresistance with EVs role and included under figure-1.

Other comments:

Comment-7: Section 5: Exosomes & CSCs. Avoid abbreviations in headings, better to write at full. Similarly, section 13: 13. ML-derived. Write ML at full.                 

 Response: Authors expanded abbreviations across side headings throughout the manuscript.

Comment-8: While authors mention the role of EVs in Chemoresistance, I would suggest authors to refer the following article (PMID: 29657282).                     

 Response: Authors mentioned the role of EVs in chemoresistance in the “EVs Role in stemness” section in the manuscript.

Comment-9: Similarly, while authors indicate the roles of EVs in Radioresistance, I recommend authors to refer the following article (PMID: 31349735).                

Response: Authors mentioned the role of EVs in radioresistance in the “EVs Role in stemness” section in the manuscript.

Minor typos:

Comment-10: Please build the abstract in a logical flow of concepts in accordance with content in the main body of the manuscript.                                                      

 Response: Authors structured and improved the abstract 

Comment-11: In the abstract: we have profoundly discussed… ! authors may wish to remove the word profoundly from the abstract. Now several journals discourage such statements.

Response: Authors eliminated the word ‘profoundly’ in the manuscript.

Comment-12: AML: write at full, when first time mentioned in the text.               

Response: Authors have expanded the abbreviation of AML in the manuscript.

Comment-13: The manuscript needs revision for typos. Few examples, but whole manuscript needs careful proof reading.                                                          

Response: We have corrected all spelling and grammatical errors throughout the manuscript accordingly and completely finished proofreading of the whole manuscript.

Comment-14: Page 2, introduction: there is extra space between [2] . Exosomes…..

 Response:  Authors corrected and removed additional space after ref. number in the manuscript.

Comment-15: - Line 111: section 2. .Exosomics                                                  

Response: Authors corrected and removed additional dot.

Comment-16: Please have proofread the manuscript for typos in the revised version.

Response: We have corrected all spelling and grammatical errors throughout the manuscript accordingly and completely finished proofreading of the whole manuscript.

Reviewer 3 Report

In this interesting paper, the Authors have reviewed the characteristic features of exosomes, nanometric vesicles generated by all types of cells, their contribution to the spread of cancer, and their utility for targeted drug and gene delivery. The Authors have focused mainly on the role of exosomes in cancer progression, metastasis, cancer cell survival, angiogenesis, and drug resistance. Highlighted is also the role of exosomes in neurodegeneration processes. The utilization of exosomes as the nanocarriers for targeted delivery of drugs, proteins, miRNAs, and peptides, has been critically evaluated and advantages of exosome nanocarriers have been assessed, including exosomes carrying phytochemicals, tumor-targeting proteins, anticancer peptides, and nucleic acids. This review presents the current state-of-the art in emerging exosome technology and is important for scientists and doctors working in cancer therapy areas, as well as for general Readership. Therefore, I recommend the paper for publication after minor revision addressing the issues listed below.

  1. The abbreviations should be defined at their first use, e.g. CSC, HNOK, EMT, SANRE,
  2. The citation in text should include only the last name, e.g. “Qadir et al. [74] concluded” instead of “Fatima Qadir et al. (2018)”; see also: Vadim Tarasov and others.
  3. In titles of sections 2 and 3, please delete the extra dot in the word “.Exosomics”.
  4. Line 62: the cited size range of exosomes of 40 to 1,000 nm seems to be larger that usually considered: 30 to 150 nm. It may include both exosomes and microvesicles, the latter being up to 1,000 nm.
  5. Recently, cancer biomarkers such as the anti-apoptotic protein of the AIP family, survivin, were detected in exosomes (e.g.: Biosensors and Bioelectronics 137 (2019) 58–71). Any comment on this subject would benefit general Readership. This relevant literature reference should be cited.
  6. Typographical and English errors:

Line 28: “resistant” – should be “resistance”;

Line 29: “advances in the role of exosomes” – change to “advances in understanding of the role of exosomes”;

Line 35: “drug resistance pathways in several cancer cells.” – should be “drug resistance pathways in several cancer cell lines.”;

Line 115: “Exosomes stimulates” – should be “Exosomes stimulate”;

Line 126: “cells mediates” – should be “cells mediate”;

Line 128: “exosomes binds” – should be “exosomes bind”;

Line 129: “exosomes can release … consequently blocks the expression” – should be “exosomes can release … consequently blocking the expression”;

Line 138: sentence starting with “A recent report by …” – there is something missing in this sentence – it is not finished; please check;

Line 152: “simultaneously infect” – should be “simultaneously infecting”;

Line 156: sentence starting with “However, a recent review” – there is something missing in this sentence and the connection with preceding sentence is lost; please check;

Line 204: “exosomes follows” – should be “exosomes follow”;

Line 208: “than” – change to “in relation to”;

Line 209: “transferring” – should be “transferrin”;

Line 220: “can mitigate inflamed brain than normal brain” – please rephrase;

Line 649: “Exosomes have extensive role” – better would be: “Exosomes play a major role”; please check;

Line 653: “Furthermore, the understanding of the contribution of exosomes in chemoresistance, CNS neurodegenerative diseases is a significant aspect to design new molecular therapies for early diagnosis and treatment optimization.” – change to, for instance: “Furthermore, better understanding of the involvement of exosomes in chemoresistance and CNS neurodegenerative diseases is a significant aspect enabling enhanced designs of novel molecular therapies for early diagnosis and treatment optimization.” Please check.

Figure 1: in the image, the words “Invagination” and “Multivesicular complex” are misspelled.

Author Response

Dear Reviewer 3:

On behalf of my coauthors, please accept my sincere thanks and gratitude for careful perusal and critical review of our manuscript entitled “The Role of Exosomes in Stemness and Neurodegenerative Diseases - Chemoresistant-Cancer Therapeutics & Phytochemicals”. We have revised the manuscript based upon the reviewers’ comments as well as your suggestion. Adequate care has been taken to accommodate each and every suggestion of the reviewers. An itemized, “point-by-point” reply to all the comments is attached separately where we have clearly presented our specific response and additions, deletions and/or modifications that have been made in the revised text, and highlighted.

Point-by-Point Answers to the Comments

Comment-1: The abbreviations should be defined at their first use, e.g. CSC, HNOK, EMT, SANRE,

Response: Authors mentioned all the abbreviations in the abbreviation list to avoid heavy text in the manuscript. 

Comment-2: The citation in text should include only the last name, e.g. “Qadir et al. [74] concluded” instead of “Fatima Qadir et al. (2018)”; see also: Vadim Tarasov and others.

Response: Authors corrected and mentioned citation author names as per the above reviewer’s suggestion in the manuscript. 

Comment-3: In titles of sections 2 and 3, please delete the extra dot in the word “.Exosomics”.

Response: Authors corrected and removed additional dot.

Comment-4: Line 62: the cited size range of exosomes of 40 to 1,000 nm seems to be larger that usually considered: 30 to 150 nm. It may include both exosomes and microvesicles, the latter being up to 1,000 nm.

Response: Authors corrected the size range of EVs and exosomes etc., and mentioned in the text of manuscript correctly and highlighted in green.

Comment-5: Recently, cancer biomarkers such as the anti-apoptotic protein of the AIP family, survivin, were detected in exosomes (e.g.: Biosensors and Bioelectronics 137 (2019) 58–71). Any comment on this subject would benefit general Readership. This relevant literature reference should be cited.

Response: Authors included the relevant literature of surviving in exosomes as a novel biomarker for early cancer detection in the section ‘EVs role in Stemness’. Authors cited the corresponding citation also.  

Typographical and English errors:

Comment-7: Line 28: “resistant” – should be “resistance”;                             

 Response: Authors have corrected this typographical error.

Comment-8: Line 29: “advances in the role of exosomes” – change to “advances in understanding of the role of exosomes”;                                                             

Response: Authors have corrected this grammar and typographical error.

Comment-9: Line 35: “drug resistance pathways in several cancer cells.” – should be “drug resistance pathways in several cancer cell lines.”;                                 

Response: Authors have corrected this grammar and typographical error.

Comment-10: Line 115: “Exosomes stimulates” – should be “Exosomes stimulate”;                                              Response: Authors have corrected this grammar and typographical error.

Comment-11: Line 126: “cells mediates” – should be “cells mediate”;              

Response: Authors corrected this grammar and typo

error.

Comment-12: Line 128: “exosomes binds” – should be “exosomes bind”;         

 Response: Authors corrected this grammar and typo

error.

Comment-13: Line 129: “exosomes can release … consequently blocks the expression” – should be “exosomes can release … consequently blocking the expression”;

Response: Authors corrected this grammar and typo

error.

Comment-14: Line 138: sentence starting with “A recent report by …” – there is something missing in this sentence – it is not finished; please check;               

Response: Authors completely mentioned the sentence in this line.

Comment-15: Line 152: “simultaneously infect” – should be “simultaneously infecting”;

 Response: Authors corrected this grammar and typo error.

Comment-16: Line 156: sentence starting with “However, a recent review” – there is something missing in this sentence and the connection with preceding sentence is lost; please check;                                                                                                        

Response: Authors corrected this grammar and typographical error.

Comment-17: Line 204: “exosomes follows” – should be “exosomes follow”;  

Response: Authors corrected this grammar and typographical error.

Comment-18: Line 208: “than” – change to “in relation to”;                             

 Response: Authors corrected this grammar and typographical error.

Comment-19: Line 209: “transferring” – should be “transferrin”;                      

 Response: Authors corrected this grammar and typographical error.

Comment-20: Line 220: “can mitigate inflamed brain than normal brain” – please rephrase;                                                                                                                         

Response: Authors corrected this grammar and typographical error.

Comment-21: Line 649: “Exosomes have extensive role” – better would be: “Exosomes play a major role”; please check;                                                                     

 Response: Authors corrected this grammar and typographical error.

Comment-22: Line 653: “Furthermore, the understanding of the contribution of exosomes in chemoresistance, CNS neurodegenerative diseases is a significant aspect to design new molecular therapies for early diagnosis and treatment optimization.” – change to, for instance: “Furthermore, better understanding of the involvement of exosomes in chemoresistance and CNS neurodegenerative diseases is a significant aspect enabling enhanced designs of novel molecular therapies for early diagnosis and treatment optimization.” Please check.                                                       Response: Authors corrected this grammar and typo error.

Comment-23: Figure 1: in the image, the words “Invagination” and “Multivesicular complex” are misspelled.                                                                                          

Response: Authors corrected this grammar and typo error in the figure-1 and replaced in the revised manuscript file.

Round 2

Reviewer 1 Report

The authors are thanked for the response. Unfortunately, there are still however inaccurate citations throughout the text. For instance the newly added ref. 3 in the first paragraph of the introduction page 3, which is a review paper by Nawaz et al. published in Stem Cells International in 2016 doi.org/10.1155/2016/1073140, does not support the stated claim: "delivery of cargo present in EVs to adjacent cancer cells induces drug resistance during chemotherapy". Indeed, such a claim better needs a research paper to back it up. However, I am not aware of any research that investigated the effects of cargo present in EVs on adjacent cancer cells drug resistance during chemotherapy. Furthermore, the same reference 3 does not perfectly fit the EV distribution based on their sizes (a few lines below). Rather, that was first stated, and more properly discussed, in a previous review paper of Nawaz et al. in Nature Review Urology in 2014 doi.org/10.1038/nrurol.2014.301. This is a major concern, because this manuscript has unusually long list of references. The authors are strongly encouraged to revisit their choices of references before considering the manuscript for publication.

Author Response

Dear Reviewer 1:

On behalf of my coauthors, please accept my sincere thanks and gratitude for careful perusal and critical review of our manuscript entitled “The Role of Exosomes in Stemness and Neurodegenerative Diseases - Chemoresistant-Cancer Therapeutics & Phytochemicals”. We have revised the manuscript based upon the your comments as well as suggestion.

Point-by-Point Answers to the Comments

Reviewer 1

Comment-1: The authors are thanked for the response. Unfortunately, there are still however inaccurate citations throughout the text. For instance the newly added ref. 3 in the first paragraph of the introduction page 3, which is a review paper by Nawaz et al. published in Stem Cells International in 2016 doi.org/10.1155/2016/1073140, does not support the stated claim: "delivery of cargo present in EVs to adjacent cancer cells induces drug resistance during chemotherapy". Indeed, such a claim better needs a research paper to back it up. However, I am not aware of any research that investigated the effects of cargo present in EVs on adjacent cancer cells drug resistance during chemotherapy. 

Response: Authors thanks to the comments and incorporated suitable corresponding references next to the text in introduction page-3. Authors request reviewer to go through the corresponding refs to support our claim as : "delivery of cargo present in EVs to adjacent cancer cells induces drug resistance during chemotherapy".  

Comment-2: Furthermore, the same reference 3 does not perfectly fit the EV distribution based on their sizes (a few lines below). Rather, that was first stated, and more properly discussed, in a previous review paper of Nawaz et al. in Nature Review Urology in 2014 doi.org/10.1038/nrurol.2014.301. This is a major concern, because this manuscript has unusually long list of references. The authors are strongly encouraged to revisit their choices of references before considering the manuscript for publication.

Response: Authors thanks to the comments and incorporated suitable corresponding references next to the text in introduction page-3. Authors performed extensive cross refs checking throughout the manuscript and eliminated some of the unnecessary refs.

Reviewer 2 Report

Authors have carefully revised their manuscript, have made the corrections and further improved the the draft. 

I endorse the publication of this review. 

However, a minor note for authors which can be incorporated during production copy. 

The review uses mix of terminologies as EVs and exosomes. i.e. in some headings it is EVs while others it is exosomes. Please use a uniform nomenclature for readers outside EV field. 

Good luck 

Author Response

Dear Reviewer  2,

On behalf of my coauthors, please accept my sincere thanks and gratitude for careful perusal and critical review of our manuscript entitled “The Role of Exosomes in Stemness and Neurodegenerative Diseases - Chemoresistant-Cancer Therapeutics & Phytochemicals”. We have revised the manuscript based upon the your  comments and  suggestion.

Reviewer 2

Comment-1: Authors have carefully revised their manuscript, have made the corrections and further improved the draft. I endorse the publication of this review.

Response:  Authors convey sincere thanks for endorsing our manuscript for publication.

Comment-2: However, a minor note for authors which can be incorporated during production copy. The review uses mix of terminologies as EVs and exosomes. i.e. in some headings it is EVs while others it is exosomes. Please use a uniform nomenclature for readers outside EV field.

Response: Authors brought uniform terminologies of exosomes at the corresponding headings throughout the manuscript.